# Aphicidal Activity and Phytotoxicity of *Citrus sinensis* Essential-Oil-Based Nano-Insecticide

**DOI:** 10.3390/insects13121150

**Published:** 2022-12-13

**Authors:** Francesca Laudani, Orlando Campolo, Roberta Caridi, Ilaria Latella, Antonino Modafferi, Vincenzo Palmeri, Agostino Sorgonà, Paolo Zoccali, Giulia Giunti

**Affiliations:** 1Department of AGRARIA, University “Mediterranea” of Reggio Calabria, Feo di Vito, 89122 Reggio Calabria, Italy; 2I.P.S.S.A.S.R.-Scigliano, Via Municipio, 87057 Cosenza, Italy; 3Department of Pharmacy, University of Salerno, Via Giovanni Paolo II 132, 84084 Fisciano, Italy

**Keywords:** *Aphis gossypii*, botanical, nano-emulsion, integrated pest management, side effect

## Abstract

**Simple Summary:**

Research about innovative sustainable ecofriendly pesticides is a key topic of global interest, aiming to reduce synthetic inputs in agriculture, to protect biodiversity, and to ensure food safety to consumers. Botanical substances, and in particular essential oils, are among the most promising natural pesticides, as can be seen from the incredibly large number of published studies in the last two decades. Nevertheless, most research is limited to laboratory studies, leaving a gap between scientific studies and field applications. In this scenario, the aim of this paper was to evaluate the feasibility of an innovative nano-insecticide containing sweet orange essential oil as the active ingredient against a key aphid pest in real conditions.

**Abstract:**

Due to its high polyphagy, *Aphis gossypii* is considered a key pest of many crops, and it can feed on hundreds of plant species belonging to the families Cucurbitaceae, Malvaceae, Solanaceae, Rutaceae, and Asteraceae. The control of this pest mainly relies on synthetic insecticides whose adverse effects on the environment and human health are encouraging researchers to explore innovative, alternative solutions. In this scenario, essential oils (EOs) could play a key role in the development of ecofriendly pesticides. In this study, the development of a citrus peel EO-based nano-formulation and its biological activity against *A. gossypii* both in the laboratory and field were described and evaluated. The phytotoxicity towards citrus plants was also assessed. The developed nano-insecticide highlighted good aphicidal activity both in the laboratory and field trials, even at moderate EO concentrations. However, the highest tested concentrations (4 and 6% of active ingredient) revealed phytotoxic effects on the photosynthetic apparatus; the side effects need to be carefully accounted for to successfully apply this control tool in field conditions.

## 1. Introduction

The widespread indiscriminate and often over-use of conventional insecticides poses several risks for the environment and human health. In addition, the ongoing emergence of insecticide resistance of several pests is undermining the range of control tools available to farmers [1]. For these reasons, both the producers and the consumers are constantly demanding pest control tools that are more ecofriendly and safer for agricultural workers and consumers compared to conventional pesticides. Among the alternatives, botanical derivatives could be the right option since many plant-synthesized active ingredients intrinsically possess a protective ability against biotic agents, including distinctive key pest species. Among botanicals, essential oils (EOs) are stimulating the attention of many researchers since the insecticidal activity of these blends of compounds is nowadays well documented [2,3]. However, despite proven bioactivity against pests, EOs present a series of characteristics (i.e., lower solubility in water, rapid degradation, flammability, and phytotoxicity) that limit their direct use in field conditions. Nevertheless, the inclusion of EOs in stable formulations could reduce their negative effects while amplifying the insecticidal action. In addition, the formulation of these substances into nano delivery systems, such as nano-emulsions or nano-particles, could also emphasize the efficacy against pests, reducing the amount of demanded active ingredients (a.i.) and, as a consequence, the impact on the environment [4].

Among the insect pests that cause the greatest agricultural production losses worldwide, *Aphis gossypii* Glover (Homoptera: Aphididae) is considered a key pest of a variety of crops, causing direct damage through both feeding on young leaves and twigs and by transmitting a series of plant viruses, including the *Citrus* tristeza virus (CTV) and the cucumber mosaic virus (CMV) [5,6]. The control of this pest is mainly based on synthetic insecticides whose massive use has resulted in the emergence of resistance phenomena that make the management of this pest even more difficult [7,8].

The aim of this experimental activity was the development of a *Citrus* peel EO-based nano-insecticide for the control of *A. gossypii,* testing its efficacy both in laboratory condition and in the field. In addition, the phytotoxic effects of the developed formulations were tested on citrus plants. The choice of using *Citrus sinensis* EO was made taking into consideration its widespread availability at a reasonable cost. Furthermore, being considered a byproduct of the orange industry, the choice of blood–sweet orange EO to develop an alternative biopesticide to control *Citrus* pests strongly relies on the general idea of a circular economy and on the specific concept of circular agriculture. On the basis of the 4R approach (reduce, reuse, recycle, and recover), circular agriculture aims to reduce external inputs by using in-house resources to improve the whole sustainability of the agroindustry and agroecosystems [9,10]. In this context, the development of a nano-emulsion containing a high concentration of a.i. enhances its feasibility in real operating conditions. Indeed, from the available literature to date, most of the proposed EO-based formulations contain quite low concentrations of EOs, posing limitations to their use in the field because of the large application volumes required for crop treatments [11].

## 2. Materials and Methods

### 2.1. Citrus Sinensis EO Extraction and Chemical Characterization

Blood–sweet orange essential oil (EO) was obtained from fruit harvested in the same orchard in which the field trials were subsequently carried out. Fresh citrus fruits were hand harvested, washed, peeled, and grinded to reduce the peels to small pieces. The EO was extracted by a traditional steam distillation apparatus (Albrigi Luigi s.r.l., Verona, Italy) and recovered in a separation funnel placed after the condenser. The recovered EO was dried over anhydrous sodium sulphate and stored at 5 °C in dark vials for further analyses and for the nano-emulsion’s formulation.

GC/MS analyses were performed with a Thermo Fisher TRACE 1300 gas chromatograph equipped with a MEGA-5 capillary column (30 m × 0.25 mm; coating thickness, 0.25 μm) and a Thermo Fisher ISQ LT ion trap mass detector (emission current: 10 microamps; count threshold: 1 count; multiplier offset: 0 volts; scan time: 1.00 s; prescan ionization time: 100 microseconds; scan mass range: 30–300 m/z; ionization mode: EI). The following analytical conditions were employed: injector and transfer line temperature at 250 and 240 °C, respectively; oven temperature programmed from 60 to 240 °C at 3 °C min^−1^; carrier gas, helium at 1 mL min^−1^; injection, 0.2 µL (10% hexane solution); split ratio, 1:30. The identification of chemicals was based on the comparison between the pure chemicals of the retention times (RTs) of the compounds and their linear retention indices (LRIs), as well as on computer matching against the commercial (NIST 05, Wiley FFNSC and ADAMS) and homemade libraries [12,13,14,15,16]. LRI was calculated by comparing the retention times of the compounds to those of a standard mixture of alkanes (C7-C30 saturated alkanes standard mixture, Supelco^®^, Bellefonte, PA, USA) [17], which was analyzed by GC/MS set at the identical conditions of the EO.

### 2.2. Oil in Water Nano-Emulsion Preparation and Characterization

The *Citrus sinensis* EO nano-emulsion (EO-NE) was prepared using the self-emulsifying process followed by sonication [3]. In detail, 15%(w) of EO was added to 5%(w) of Tween^®^ 80 (HLB value = 15), and the obtained mixture was stirred for 30 min at 10,000 RPM; then, distilled water (80% w) was added drop wise (1 mL min^−1^). This coarse emulsion was stirred for three hours at 10,000 RPM and sonicated for 3 subsequent cycles (1.5 min) by means of an UP200ST ultrasonic immersion homogenizer (Hielsher^©^, Teltow, Germany) at 100 W power. In order to avoid the heating of the emulsion, which can cause the degradation of the EO, the sonication process was carried out in an ice bath.

The average droplet size and size distribution (polydispersity index—PDI) were measured by using a dynamic light scattering (DLS) particle size analyzer (Z-sizer Nano, Malvern Instruments) at 25 °C. In addition, the particle charge was quantified as zeta potential using a Z-Sizer nano (Malvern Instruments) at 25 °C. To attain the correct measurements from the instrument, 0.5 mL of EO-NE was diluted in 100 mL of double-distilled water, and the aliquots (1 mL for droplet dimension and 0.75 mL for droplet surface charge) of the diluted emulsion were analyzed. The physical characteristics of EO-NE were tested after 24 h from the preparation and monthly for one year. Three replicates of fourteen cycles were provided for the tested sample. Three samples were analyzed as replicates for every tested measure.

### 2.3. Aphis Gossypii Rearing

The original cotton aphid colony came from an organic citrus crop (cv. Tarocco) placed in Calabria (southern Italy). The aphid specimens used in laboratory experiments were reared on young zucchini plants grown in a 36 m^3^ walk-in climatic chamber (Cavallo srl, Buccinasco, Italy) kept at 26 °C and 70% RH. Lighting (16:8, light: dark photoperiod) was provided by ten 200 W halogen lamps (approx. 333 W (m^2^)^−1^. In order to obtain enough *A. gossypii* specimens for the trials, weekly noninfested zucchini plants were placed inside aphid breeding cages (Bugdorm^®^) and put in contact with infested leaves to allow aphid spread and reproduction. Only mature apterous females were used in the laboratory trials. Every insect was used only once.

### 2.4. EO-NE Toxicity against A. Gossypii

#### 2.4.1. Laboratory Trials

To evaluate the efficacy of the developed formulation, the zucchini seedlings (each provided with 4 well-expanded leaves) were infested with 30 specimens, and then, after they were fixed to the leaf surface (1 h approx.), the infested plants were sprayed until runoff using a 2 L power-pack aerosol hand sprayer (Dea^®^, Volpi, Italy) with the EO-NE solutions at 1, 2, 4, and 6 g × 100 mL^−1^ of distilled water. Negative control treatments were carried out with water alone, whereas deltamethrin (DECIS^®^ EVO, Bayer Crop Science) applied at the maximum recommended rate for citrus aphids (50 mL hL^−1^) was used as the positive control.

Mortality was assessed 24, 36, and 48 h after the beginning of the experiments. Specimens were considered dead when they remained immobile after being stimulated with a fine paintbrush. Each treatment was replicated 5 times. 

#### 2.4.2. Field Trials

The field trials were carried out during spring 2021 in a 4-hectare citrus orchard under integrated pest management located at Locri (province of Reggio Calabria, Italy) (38°14′50.2″ N 16°16′10.2″ E) at 20 m above the sea level, where no chemical treatments were applied one year before the trials. The trees were 30-year-old orange trees (cv. Tarocco nucellare) planted in a 5 × 6 m grid. The experiments were carried out in spring when the young shoots were highly infested by aphids. Preliminary investigations were carried out a few days before the beginning of the experiments to determine the aphid population density and specificity (>95% belonging to the target specie *A. gossypii*). The different treatments were applied to citrus trees using a 5 L power-pack aerosol hand sprayer (Eurospin^®^, Italy). The trees were sprayed until runoff with EO-NE solutions at different concentrations, using the same application rates carried out in the laboratory trial (i.e., 1, 2, 4, and 6 g × 100 mL^−1^ of distilled water). Both positive (deltamethrin) and negative controls (water) were performed with the same application methods used for the EO-NE solutions. 

For each EO-NE dilution and treatment, four randomly selected trees were sprayed in a completely randomized model. Aphid mortality was assessed for each tree on four previously labeled infested shoots after the same time intervals carried out in the laboratory experiment (i.e., 24, 36, and 48 h from the treatments). 

The field treatment efficacy was calculated as follows: (1)$$Field\;treatment\;efficacy=\frac{\left(R_{treatment}-R_{control}\right)}{\left(100-R_{control}\right)}\times 100$$
where *R* represents the reduction rate due to the treatments calculated as follows:(2)$$R=\frac{\left(P_{before\;treatment}-P_{after\;treatment}\right)}{P_{before\;treatment}}\times 100$$
where *P* represents the population (number of alive specimens) registered in each sample.

### 2.5. Phytotoxicity Analysis

Phytotoxicity analyses were carried out by photosynthetic rate measurements as a sensible physiological trait for different reasons: a non-destructive and quick method, the best ecotoxicological method [18], and an alternative to growth inhibition test [19].

The photosynthetic rate measurements were carried out on at least three healthy leaves of five orange trees treated with EO-NE solutions at the same application rates used in the laboratory trials (i.e., 1, 2, 4, and 6 g × 100 mL^−1^ of distilled water) or with the negative control (water alone). 

A calibrated portable photosynthesis system (LI-6400; LI-COR, Inc.; Lincoln, NE) was used to measure the photosynthetic rate (μmol (CO_2_) m^–2^ s^–1^) at 500 cm^3^ min^−1^ of flow rate, 26 °C of leaf temperature, 400 μmol(CO_2_) mol(air)^–1^ of CO_2_ concentration (controlled by CO2cylinder), and 1200 µmol m^−2^s^−1^ of photosynthetically active radiation supplied by the LED light source in the leaf chamber. Each measurement was made with a minimum and maximum wait time of 120 and 200 s, respectively, and by matching the infrared gas analyzers to a 50 μmol (CO_2_) mol(air)^–1^ difference in CO^2^ concentration between the sample and the reference before every change of plants. The leaf-to-air vapor pressure difference (VPD) was set to 1.5 kPa and adjusted at a constant level by manipulating the humidity of incoming air as needed.

### 2.6. Data Analysis 

Dependent variables were tested for homogeneity and normality of variance (Levene and Shapiro–Wilk tests, respectively), and because they met the ANOVA assumptions (*p* < 0.05), no data transformation was required.

The laboratory efficacy of the tested formulations was corrected for negative control mortality using Abbott’s formula [20]. Mortality was subjected to the bivariate full factorial analysis of variance (ANOVA) procedure, and multiple comparisons among the variables (i.e., concentration, time, and concentration × time) were performed using Tukey’s HSD (honestly significant difference) post hoc test. Robust standard errors were calculated using the HC method.

Probit analysis was performed in order to estimate the median lethal concentrations (LC_50_ and LC_90_) for laboratory trials. The lethal concentration values were considered significantly different if their 95% fiducial limits did not overlap.

The photosynthetic rate results were analyzed by two-way ANOVA with the treatment and time as the main factors with different levels (CTR, 1, 2, 4, and 6% for the concentration; 0, 1, and, 6 days for the time) and concentration–time interaction. Then, Tukey’s HSD post hoc test was used to compare the means of the photosynthetic rate data of each treatment within each time. 

All statistical analyses were performed using the software SPSS v. 22 (IBM, Armonk, NY, USA).

## 3. Results

### 3.1. Citrus Sinensis EO Chemical Characterization and Nano-Emulsion Characterization

A total of 29 different compounds were detected in the blood–sweet orange EO. D-limonene was the most abundant compound (93.35%) detected, followed by β-myrcene (3.38%), and α-pinene (1.14%), whereas the abundance of the other 26 compounds was lower than 1%. Only three compounds (α-cubebene, γ-muurolene, and caryophyllene oxide) were only in traces (an abundance less than 0.01%) (Table 1). 

Monoterpene hydrocarbons (98.59%) represented the great majority of the detected chemicals, while oxygenated monoterpenes, sesquiterpene hydrocarbons, oxygenated sesquiterpenes, and non-terpene aldehyde ranged from 0.73 to 0.01%. The developed nano-emulsion had particle dimensions of 131.36 ± 0.50 nm with a polydispersity index (PDI) of 0.11 and a surface charge (ξ potential) of −23.76 ± 0.75 mV.

### 3.2. Aphicidal Activity

In the laboratory trials, the aphid mortality registered for the negative control after 24, 36, and 48 h from the treatment was 0, 1.6, and 4%, respectively, whereas the positive control (deltamethrin) caused the complete mortality of all the exposed insects immediately after 24 h from the treatment. In all the time intervals (24, 36, and 48 h after the treatments) the mortality induced by the EO-NE treatments fitted (*p* > 0.05) with the Probit model, highlighting a dose-dependent response. Twenty-four hours after the insecticide treatments, both the LC_50_ and LC_90_ values were higher (2.27 and 4.35%, respectively) than those recorded at 36 h (LC_50_ = 1.88% and LC_90_ = 3.56%) and 48 h after the insecticide treatments (LC_50_ = 1.48% and LC_90_ = 2.86%) (Table 2).

Both the EO-NE concentration as well as the time after the treatment significantly affected aphid mortality (concentration: F = 464.05; df = 3; *p* < 0.01; time: F = 10.43; df = 2; *p* < 0.01), whereas their interaction did not affect the efficacy (F = 1.328; df = 6; *p* > 0.05). The lowest tested concentration (i.e., 1% of EO in the nano-emulsion) was able to kill less than 15% of the exposed insects. Conversely, the mortality caused by nano-emulsions containing 2, 4, and 6% of EO ranged from 38.67 to 100%, with the two highest doses (i.e., 4 and 6%) provoking the death of almost all tested insects just after 24 h (mortality = 89.33 ± 3.86 and 96 ± 2.67%, respectively). In detail, the insecticidal activities recorded for 4 and 6% EO-NE were similar without accountable statistical differences between the two dilutions (*p* > 0.05) (Figure 1).

In the field, the efficacy (reported as field treatment efficacy) of the developed nano-insecticide showed a similar trend compared to that described in the laboratory trials (Figure 2). The concentration used, as well as the time between the treatment and the sampling, had a significant effect on aphid mortality (concentration: F = 505.793; df = 3; *p* < 0.01; time: F = 22.008; df = 2; *p* < 0.01); however, similar to laboratory trials, their interaction (concentration × time) did not determine statistical differences (*p* = 0.05). Indeed, the mortality induced by the two highest tested concentrations was not significantly different, regardless of time.

The recorded mortality, 24 h after the treatments, ranged from 4.88 ± 6.16 to 89.39 ± 6.34% depending on the EO-NE concentration. In the sampling carried out 36 h after the treatments, the aphicidal activity increased, and at the two highest application rates (i.e., 4 and 6%), the efficacy reached 85.68 ± 4.66 and 91.85 ± 5.16%, respectively. Lastly, 48 h after the treatments, aphid mortality on the plants treated with the EO-NE containing 4 and 6% of EO was in both cases higher than 90%. The positive control (deltamethrin) caused higher mortality rates than those recorded for the EO-treated plants; indeed, after 24 h no aphids were recorded alive in the samples treated with deltamethrin.

### 3.3. Phytotoxicity

The highest EO-NE concentrations (i.e., 4 and 6%) caused toxic effect on the photosynthetic apparatus of citrus plants. One day after treatment, the photosynthetic rate was lower in the plants treated with EO-NE applied at 4 and 6% of a.i. than in the other tested concentrations and the water control (concentration–time interaction: F = 2.8274; df = 8, *p* < 0.01). The photosynthetic rate was reduced by 52% and 51% in plants treated with 4 and 6% of a.i., respectively, compared to the water control. The phytotoxic effects induced by the two highest concentrations was observed up to six days after treatment, while in the plants treated with EO-NE at 1 and 2% of a.i., no phytotoxic effects were highlighted throughout the trial (Figure 3).

## 4. Discussion

The compounds detected in *C. sinensis* EO were almost totally represented by monoterpene hydrocarbons, and among them, D-limonene accounted for nearly the entire blend of the EO. The chemical composition of *C. sinensis* EO can be influenced by different seasonal and climatic parameters, as well as the ripening stage, cultivar, and growing localities [22]. The development of insecticide formulations can mitigate the negative effects of crude EOs, such as rapid degradation, flammability, low solubility in water and phytotoxicity. The biological activity of EOs was verified against several pests and vectors [23,24,25], while the side effects towards non-target organisms strongly depend on the non-target species, as well as the EO used as insecticide [2,4,26].

Eos presenting insecticidal activity against target pests usually affect different biological pathways, mainly related to the nervous system of insects (i.e., octopamine receptor, GABA channel, and ACHE activity) [27]. The developed EO-NE presented interesting physical characteristics related to the small droplet dimensions (<130 nm) and the homogeneity of the produced droplets (i.e., polydispersity index near to zero). In EO-based formulations, the small particles ranging in the nano-scale dimensions generally improve both the persistence and effectiveness of a.i. against insect pests (de Oliveira et al., 2014); therefore, the low required amount of a.i. makes these nano-formulations even economically sustainable. Specifically, among the EOs known for their insecticidal activity, those extracted from *Citrus* spp. fruit peels are promising sources for the development of eco-friendly control tools, mostly because of their large availability on the market and their generally reasonable cost [3].

To the best of our knowledge, so far, the evaluation of EO-based insecticides against the cotton aphid has been conducted exclusively under laboratory conditions; thus, their evaluation under field operating conditions was lacking. In this scenario, the present study firstly demonstrated the efficacy of these insecticide formulations also in the field. The field trials highlighted that EO-NE caused a field treatment efficacy comparable to the positive control, deltamethrin, after 48 h from the application. Most of the previous studies about aphid species performed fumigation trials to determine the insecticidal activity of the EOs toward the target pests (e.g., [28,29,30]); thus, due to the deep differences among the experimental set-ups, the comparison between our results and those reported in the literature is quite unreliable and unrealistic. Nevertheless, the successful application of nanotechnology for the control of *A. gossypii* seems to be promising to also improve the safety and effectiveness of other plant-borne a.i., optimizing their commercial formulations; as an example, nano-formulated pyrethrins were harmless, in terms of mortality and longevity, to the predators *Coccinella septempunctata* L. (Coleoptera: Coccinellidae) and *Macrolophus pygmaeus* Rambur (Hemiptera: Miridae), respectively, while they show superior toxicity against the aphid [31].

Concerning crops, the ecotoxicological tests of pesticides are based on growth inhibition and/or visualization symptoms on representative non-target plant species [32]. However, as a physiological endpoint for the in vivo phytotoxic test, the photosynthesis rate was used as a reliable nondestructive indicator of plant status during the exposure to pesticides and chemicals [33]. Moreover, it was also demonstrated that the inhibition of photosynthetic machinery is a putative mode of action of EOs [34] Indeed, it is well known that the photosynthetic rate is “…*a plant-driven response to the perception of stress rather than a secondary physiological response to tissue damage*…”, highlighting strict interactions between photosynthesis, ROS, and hormonal signaling pathways for plant responses to stress [35]. The results from phytotoxicity trials that indicated a negative effect from the highest EO application rates on the photosynthesis of orange leaves, which was observed within one day of treatment. Different mechanisms were proposed for the photosynthesis inhibition driven by EOs, including the reduction of photosynthesis pigments (chlorophylls and carotenoids) [36] and the degradation of PSII [37]. A key point highlighted by the experimentation was the maintenance of the EO-induced toxic effect during the experimental time-course, suggesting the lack of plant detoxification mechanisms for this EO, which were previously observed in other plants treated with diverse EOs [34]. The results suggest that the photosynthetic apparatus of orange leaves was deeply damaged by the EO treatments.

## 5. Conclusions

During the development and evaluation of new control tools against pests, a series of variables must be considered to achieve potentially transferable results in real operative conditions. As an example, despite the fact that many studies have evaluated the pesticidal activity of EOs, only a few papers also took into account the undesirable effects toward non-target species, including crop plants. In this scenario, deeper knowledge about the ecotoxicological impact and environmental fate of botanical-based pesticides is still needed.

In this paper, the aphicidal potential of the developed *Citrus* EO nano-formulation was demonstrated, although further improvements are auspicial to reduce the phytotoxic effects at the highest application rates towards the crops. Nevertheless, lower EO concentrations were safe for the citrus trees and demonstrated good insecticidal activity against *A. gossypii,* suggesting that a compromise between the release rate and the concentration of the a.i. can be the key strategy to overcome the negative effects on the crop granting effective pest control.

## Figures and Tables

**Figure 1 insects-13-01150-f001:**
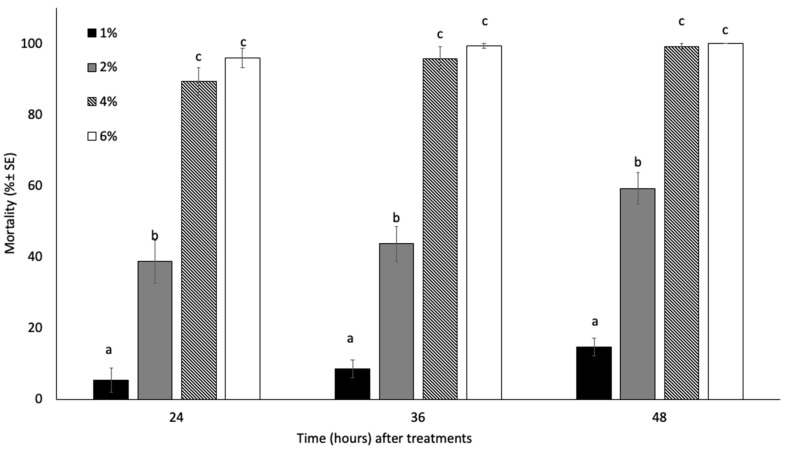
Mean percent mortality ± SE of *A. gossypii* adults exposed to different concentrations of *Citrus sinensis* EO nano-emulsion after 24, 36, and 48 h in laboratory trials. Different letters indicate statistical differences among the different treatments at the same exposure time (ANOVA, *p* < 0.05).

**Figure 2 insects-13-01150-f002:**
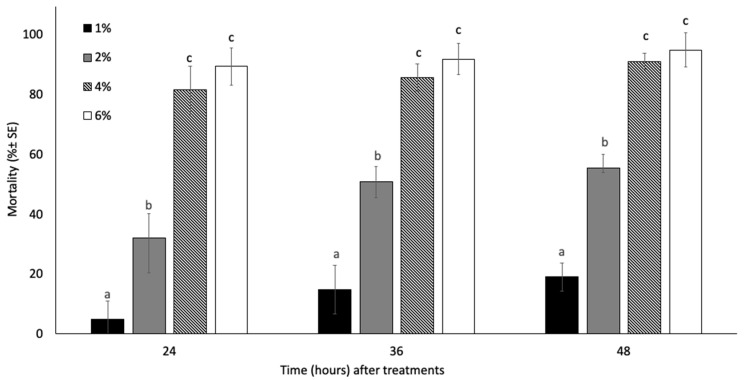
Mean percent mortality ± SE of *A. gossypii* adults exposed to different concentrations of *Citrus sinensis* EO nano-emulsion after 24, 36 and 48 h in field trials. Different letters indicate statistical differences among the different treatments at the same exposure time (ANOVA, *p* < 0.05).

**Figure 3 insects-13-01150-f003:**
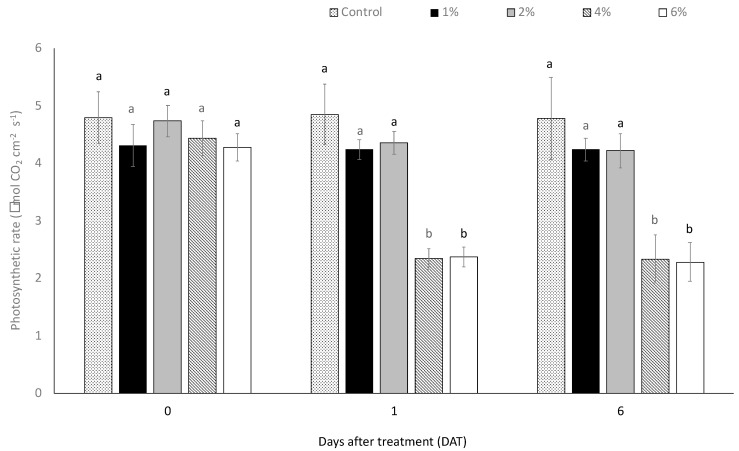
Photosynthetic rate (µmol CO_2_ cm^−2^ s^−1^) ± SE registered in the plants at 0, 1, and 6 days after treatments with different concentrations of *Citrus sinensis* EO nano-emulsion. Different letters indicate statistical differences among the different treatments at the same exposure time (ANOVA, *p* < 0.05).

**Table 1 insects-13-01150-t001:** Chemical composition of the essential oil extracted from sweet orange fruit (cv. Tarocco nucellare).

Compound ^a^	LRI Exp ^b^	LRI Literature ^c^	Retention Time (min)	Relative Peak Area (%)
α-thujene	928	931	6.52	0.01
α-pinene	935	939	6.70	1.14
Camphene	950	953	7.11	0.01
Sabinene	975	976	7.77	0.50
β-pinene	979	980	7.89	0.03
β-myrcene	993	991	8.24	3.38
Octanal	1007	1001	8.68	0.33
δ-3-Carene	1013	1011	8.90	0.18
D-limonene	1033	1031	9.64	93.35
Terpinolene	1090	1088	11.68	0.02
Linalool	1102	1098	12.11	0.54
Trans-p-mentha-2,8-dienol	1123	1118	13.05	0.02
Cis-limonene oxide	1133	1134	13.46	0.04
Trans-limonene oxide	1138	1139	13.66	0.04
Citronellal	1155	1153	14.42	0.03
α-terpineol	1192	1189	16.01	0.07
Decanal	1212	1208	16.85	0.10
α-cubebene	1356	1351	23.11	tr ^d^
α-copaene	1367	1376	23.57	0.04
β-cubebene	1381	1390	24.17	0.04
Z-β-caryophyllene	1410	1406	25.38	0.02
E-β-caryophyllene	1420	1418	25.78	0.04
α-caryophyllene	1444	1454	26.78	0.01
γ-muurolene	1468	1477	27.73	tr
Germacrene D	1473	1481	27.92	0.02
Valencene	1484	1491	28.40	0.02
α-muurolene	1492	1499	28.71	0.01
δ-cadinene	1515	1524	29.61	0.04
Caryophyllene oxide	1578	1581	32.02	tr
Class compound				Relative abundance
Monoterpene hydrocarbons				98.59
Oxygenated monoterpenes				0.73
Aldehydes				0.43
Sesquiterpene hydrocarbons				0.24
Oxygenated sesquiterpenes				0.01

^a^ Compounds are listed in order of their retention times from a MEGA-5 column. ^b^ Linear retention index on MEGA-5 column experimentally determined using homologous series of C7-C30 alkanes. ^c^ Linear retention index taken from Adams ([12], or NIST 05 [21]) and the literature. ^d^ Traces = % below 0.01.

**Table 2 insects-13-01150-t002:** Toxicity of sweet orange essential oil nano-emulsion against *Aphis gossypii*.

Time (h after Treatment)	LC_50_ ^a^ (g hg^−1^–95%FL ^b^)	LC_90_ ^a^ (g hg^−1^–95%FL)	Χ^2 c^	Significance
24	2.27 (1.92–2.65)	4.35 (3.61–5.76)	0.364	ns ^d^
36	1.88 (1.59–2.21)	3.56 (2.94–4.79)	1.478	ns
48	1.48 (1.22–1.74)	2.86 (2.34–4.03)	1.607	ns
Significance	*	*		

^a^ LC_50_ and LC_90_: lethal concentration that killed 50 and 90% of the treated insects, respectively. ^b^ FL: fiducial limit. ^c^ Χ^2^: Pearson’s goodness-of-fit test of dose–mortality response. ^d^ ns: not significant. *: *p* < 0.05.

## Data Availability

The data will be available from the corresponding author upon reasonable request.

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
