# Peer review of "Aphicidal Activity and Phytotoxicity of Citrus sinensis Essential-Oil-Based Nano-Insecticide"

_insects, 2022, doi:10.3390/insects13121150_

Round 1
Reviewer 1 Report
The authors present a nice experiment about the use of essential oils on aphids in citrus crops. Organization and information presented is nicely done. A few minor comments below.
Specific comments:
Line 126: Should be spread instead of widespread
Line 139: Should stimulated instead of stimulate
Lin 167: No-destructive should be non-destructive
Line 184-185: Make sure you indicate here or in your results which variables you transformed.
Line 203: Was detected should be were detected.
Table 1: Maybe you need a new header when you get into the Monoterpene Hydrocarbons section to differentiate it a little more
Line 237: Check the interaction p-value. Do you mean p > 0.05 if not significant?
Line 296: What is PDI?
Line 342: Phrasing is odd here.
Author Response
REPLY TO REVIEWER 1
Reviewer 1: The authors present a nice experiment about the use of essential oils on aphids in citrus crops. Organization and information presented is nicely done. A few minor comments below.
Authors: We would like to thank Reviewer 1 for the kind appreciation and the useful comments. Please find below a point-to-point reply to specific comments.
Specific comments:
Reviewer 1: Line 126: Should be spread instead of widespread
Authors: Done
Reviewer 1: Line 139: Should stimulated instead of stimulate
Authors: Done
Reviewer 1: Lin 167: No-destructive should be non-destructive
Authors: Done
Reviewer 1: Line 184-185: Make sure you indicate here or in your results which variables you transformed.
Authors: Data were processed without transformation since they met the assumptions of the ANOVA (i.e., homoscedasticity and normality). The sentence in M&M was modified accordingly. Thank you for your useful comment that make us note this important refuse. Done
Reviewer 1: Line 203: Was detected should be were detected.
Authors: Done
Reviewer 1: Table 1: Maybe you need a new header when you get into the Monoterpene Hydrocarbons section to differentiate it a little more
Authors: Done
Reviewer 1: Line 237: Check the interaction p-value. Do you mean p > 0.05 if not significant?
Authors: This was a typo. The symbol was corrected. Done
Reviewer 1: Line 296: What is PDI?
Authors: Thanks for the recommendation. Indeed, we forgotten to indicate the meaning of the acronym. Explanation was also included in M&M section. Done
Reviewer 1: Line 342: Phrasing is odd here.
Authors: This section has been revised and the sentence rephrased. Done
Reviewer 2 Report
This article by Laudani et al. discusses the insecticidal and phytotoxic effects of nano-formulated orange essential oil. The article is very clear, well written and the experimental design is simple and well constructed. The work is interesting since it evaluates the feasibility of field application of this natural and low cost insecticide.
I have no major remarks to oppose the publication of this article. My main concerns go to the statistics that need corrections and explanations (see remarks below). The bibliography needs to be reworked in its form (see remarks below).
Below, my remarks line by line:
L139: add “y” to “the”
L185: data were transformed with the arcsin function. I guess it is for mortality (expressed as percentage) to normalize the data but why then using glm?
L188: the authors don’t use here classical survival models. They should explain why. They pretend to use glm models that are family of models. What would be informative is to say which law is taken to describe the mortality in the error model of the glm.
L192. Here again the comparison of LC using probit, seems old fashion. Why not using survival curves and estimation of LC50 using the 3 time points? It might be usual practices in toxicology, but then the authors should justify these choices.
L213: cite correctly Adams and NIST 05
L230: italicize species name
L230: This approach is not familiar to me and at least it should be explained what are being tested by line and by column in table 2. Even by reading the text, the line and column “Sign” are not clear.
L237: The sentence needs to be clarified (the link between interaction and efficacy)
L246 (and statistical analysis): Why the authors made the choice to consider time as a qualitative factor rather than a quantitative one? To my opinion lines (or curves) between the 3 time points would be much more explicit than the bar-chart proposed (that at least should be replaced by box plots or violin plots).
L246 (256 and 278): how are the SE calculated? Are the classical SE calculated independently of the ANVOVA2 (if so it is incorrect)? Are they calculated using the law introduced in the glm model? Are they calculated by reverse transformation of the IC from the arcsin? This point highlights the need of better explanations of the statistical analyses performed.
L253: here again the interpretation of interaction is not clear even though I understand that interaction is not important here as only linked with the Y data that is bounded at 100%.
L298 (321): refs to be cited correctly
L300: double space?
L364: most species names are not italicized, journal names (and authors) are sometimes capitalicised (ref 15 and 16 seems incomplete). This part (References) should be corrected.
Author Response
REPLY TO REVIEWER 2
Reviewer 2: This article by Laudani et al. discusses the insecticidal and phytotoxic effects of nano-formulated orange essential oil. The article is very clear, well written and the experimental design is simple and well constructed. The work is interesting since it evaluates the feasibility of field application of this natural and low cost insecticide.
I have no major remarks to oppose the publication of this article. My main concerns go to the statistics that need corrections and explanations (see remarks below). The bibliography needs to be reworked in its form (see remarks below).
Authors: We would like to thank Reviewer 2 for the nice opinion about our paper. We have carefully addressed all suggested corrections and comments, and we have reply in detail to every single point raised by Reviewer 2. Although we cannot always totally agree with Reviewer 2 opinion, we provide replies to make understandable our choices and the approach used in the experimental set up. We hope our replies can be found satisfactory and acceptable.
Below, my remarks line by line:
Reviewer 2: L139: add “y” to “the”
Authors: Done
Reviewer 2: L185: data were transformed with the arcsin function. I guess it is for mortality (expressed as percentage) to normalize the data but why then using glm?
Authors: Data were processed without transformation since they met the assumptions of the ANOVA (i.e., homoscedasticity and normality). The sentence in M&M was modified accordingly. Thanks for your useful comment that highlighted this refuse in the data analysis section. Done
Reviewer 2: L188: the authors don’t use here classical survival models. They should explain why. They pretend to use glm models that are family of models. What would be informative is to say which law is taken to describe the mortality in the error model of the glm.
Authors: We are sorry, the statistical methods could be misunderstood in our previous version. There was also a refuse about data transformation. In detail, we subjected the data to a bivariate analysis of variance (ANOVA) which, in the spss software, is included among the general linear models (not generalized). However, we decide to rephrase this part to improve clarity.
Concerning the use of survival curves, in our opinion, the classical survival models are very useful tools in ecotoxicology and in medicine, whereas in the evaluation of insecticide efficacy the probit model is preferred since it is considered a standard method to evaluate the efficacy of tested a.i.. Please, see also the reply to next comment. Done
Reviewer 2: L192. Here again the comparison of LC using probit, seems old fashion. Why not using survival curves and estimation of LC50 using the 3 time points? It might be usual practices in toxicology, but then the authors should justify these choices.
Authors: We partially agree with the reviewer. From an eco-toxicological point of view the mortality (or survival) curves are widely used since the objectives may be different compared to our approach. In studies aimed at evaluating the efficacy of insecticides against pests, the probit model is widely used since it allows to highlight, measure and estimate not only the efficacy but also any insecticide resistance phenomena. In addition, lethal concentrations (or doses) are used to compare the susceptibility of a specific pest to different a.i.; thus, we prefer to maintain this approach since the results of our experiments may have implications in real pest control.
Furthermore, considering our experimental set up, survival models would fit worst to our dataset. Indeed, the mortality of insect specimens was checked just for acute toxicity and just for 3 exposure time; the observations lasted 48 hours, which should not enough to provide reliable survival curves. Furthermore, survival models generally consider as “replicate” the single specimen tested; here, in contrast, we use cohort of insects both in laboratory and in field conditions, where this approach is almost impossible to achieve.
Reviewer 2: L213: cite correctly Adams and NIST 05
Authors: Done
Reviewer 2: L230: italicize species name
Authors: Done
Reviewer 2: L230: This approach is not familiar to me and at least it should be explained what are being tested by line and by column in table 2. Even by reading the text, the line and column “Sign” are not clear.
Authors: “Sign” meant Significance. In the last column, the significances of the Pearson’s goodness of fit tests were reported. N.s. (not significant) means that the probit model fits the experimental data. The term “Sign” in the last row indicates the statistical differences among the LC values when their 95% fiducial limits did not overlap. The explanation was already reported in Data analysis section, but we decided to change the headlines in “Significance”. Done
Reviewer 2: L237: The sentence needs to be clarified (the link between interaction and efficacy)
Authors: Thanks for this comment. As highlighted by Reviewer 1, here there was a typo in the p value. We have amended the sentence and we believe that now this point is clear. Done
Reviewer 2: L246 (and statistical analysis): Why the authors made the choice to consider time as a qualitative factor rather than a quantitative one? To my opinion lines (or curves) between the 3 time points would be much more explicit than the bar-chart proposed (that at least should be replaced by box plots or violin plots).
Authors: We agree that lines are more indicate than bars to follow the trend of a phenomenon. In the specific case, however, the aspects that are important to highlight are the differences in mortality among the different concentrations. Indeed, from a practical point of view it is more useful to know the final mortality induced by the formulation developed rather than the time needed to obtain a specific mortality. Furthermore, the exposure times considered here are close because our intent was to check mainly acute toxicity rather than lethal times. The mortality is expected to increase because we consider the cumulative mortality at each considered time. Since we used short intervals between the mortality evaluations, we decided to consider time as fixed factor rather than quantitative factor.
Reviewer 2: L246 (256 and 278): how are the SE calculated? Are the classical SE calculated independently of the ANVOVA2 (if so it is incorrect)? Are they calculated using the law introduced in the glm model? Are they calculated by reverse transformation of the IC from the arcsin? This point highlights the need of better explanations of the statistical analyses performed.
Authors: The standard errors reported in the graphs and text were calculated by spss when running the ANOVA analyses (robust SE with HC method). Please, consider our previous reply about glm and transformation of the data. Therefore, the sentence in Data analysis section was rephrased. Done
Reviewer 2: L253: here again the interpretation of interaction is not clear even though I understand that interaction is not important here as only linked with the Y data that is bounded at 100%.
Authors: Done
Reviewer 2: L298 (321): refs to be cited correctly
Authors: Done
Reviewer 2: L300: double space?
Authors: Done
Reviewer 2: L364: most species names are not italicized, journal names (and authors) are sometimes capitalicised (ref 15 and 16 seems incomplete). This part (References) should be corrected.
Authors: Done